# Influence of Bioactive Compounds Incorporated in a Nanoemulsion as Coating on Avocado Fruits (*Persea americana*) during Postharvest Storage: Antioxidant Activity, Physicochemical Changes and Structural Evaluation

**DOI:** 10.3390/antiox8100500

**Published:** 2019-10-21

**Authors:** Antonio de Jesus Cenobio-Galindo, Juan Ocampo-López, Abigail Reyes-Munguía, María Luisa Carrillo-Inungaray, Maria Cawood, Gabriela Medina-Pérez, Fabián Fernández-Luqueño, Rafael Germán Campos-Montiel

**Affiliations:** 1Instituto de Ciencias Agropecuarias, Universidad Autónoma del Estado de Hidalgo, Av. Rancho Universitario s/n Km. 1., Tulancingo C.P. 43600, Hidalgo, Mexico; anjec_hs@hotmail.com (A.d.J.C.-G.); jocampo@uaeh.edu.mx (J.O.-L.); gamepe@yahoo.com (G.M.-P.); 2Unidad Académica Multidisciplinaria Zona Huasteca, Universidad Autónoma de San Luis Potosí, Romualdo del campo No. 501, Fracc. Rafael Curiel, C.P. Ciudad Valles, SLP C.P. 79060, Mexico; aby1974@hotmail.com (A.R.-M.); maluisa@uaslp.mx (M.L.C.-I.); 3Department of Plant Sciences, University of the Free State, Bloemfontein 9301, South Africa; CawoodME@ufs.ac.za; 4Transdisciplinary Doctoral Program in Scientific and Technological Development for the Society, Cinvestav-Zacatenco, Mexico City C. P. 07360, Mexico; 5Sustainability of Natural Resources and Energy Program, Cinvestav-Saltillo, Coahuila de Zaragoza C. P. 25900, Mexico; fabian.fernandez@cinvestav.edu.mx

**Keywords:** antioxidants, encapsulation, orange essential oil, xoconostle, maturation

## Abstract

The objective of the present study was to determine the effect of the application of a nanoemulsion made of orange essential oil and *Opuntia oligacantha* extract on avocado quality during postharvest. The nanoemulsion was applied as a coating in whole fruits, and the following treatments were assessed: concentrated nanoemulsion (CN), 50% nanoemulsion (N50), 25% nanoemulsion (N25) and control (C). Weight loss, firmness, polyphenol oxidase (PPO) activity, total soluble solids, pH, external and internal colour, total phenols, total flavonoids, antioxidant activity by 2,2′-Azino-bis(3-ethylbenzothiazoline-6-sulfonic acid) (ABTS) and 2,2-diphenyl-1-picrylhydrazyl (DPPH), while the structural evaluation of the epicarp was assessed through histological cuts. Significant differences were found (*p* < 0.05) among the treatments in all the response variables. The best results were with the N50 and N25 treatments for firmness and weight loss, finding that the activity of the PPO was diminished, and a delay in the darkening was observed in the coated fruits. Furthermore, the nanoemulsion treatments maintained the total phenol and total flavonoid contents and potentiated antioxidant activity at 60 days. This histological study showed that the nanoemulsion has a delaying effect on the maturation of the epicarp. The results indicate that using this nanoemulsion as a coating is an effective alternative to improve the postharvest life of avocado.

## 1. Introduction

Avocado (*Persea americana*) is native to central and southern Mexico but consumed globally [1]. The main problem for avocado growers, marketers and consumers is the short shelf life of avocados. Therefore, it is necessary to develop natural environmentally friendly products to improve the shelf life of avocados [2].

Xoconostle (*Opuntia oligacantha*) is a fruit from the central area of Mexico. Several reports have indicated that this fruit contains bioactive compounds such as phenolic compounds with antioxidant and antimicrobial properties [3]. However, these bioactive compounds are affected by pH, oxygen, and exposure to light or temperature, limiting their use in the food industry [4]. Cenobio-Galindo et al. [5] encapsulated *Opuntia oligacantha* extract and then incorporated it into starch films, finding that the encapsulated bioactive compounds were protected and available to act when released.

Orange essential oil (*Citrus sinensis*) is one of the most used essential oils in various industries, such as the food industry. It contains volatile monoterpenes with limonene identified as the major constituent, but its low solubility, volatility and instability during processing and storage makes it difficult to use. Therefore, it is necessary to incorporate this oil in a stable system that allows it to maintain its characteristics [6,7]. 

Nanoemulsions are encapsulation systems with characteristics of small particles (*r* < 100 nm), allowing excellent stability against separation and aggregation [8] and they also exhibit increased bioavailability of the encapsulated ingredients [9]. A potential advantage of these dispersions is that they have high compatibility with the components of foods and can even cross biological membranes without difficulty [8,10]. Nanoemulsions have been used mainly in the development of some foods and beverages to contribute to the stability of the foods that contain them [11]. Zambrano-Zaragoza et al. [12] studied the effect of a coating with α-tocopherol and cactus mucilage that were applied on apples and found that the nanoemulsion helped to maintain the firmness of the apples and decreased the activity of certain enzymes in the fruits. Oh et al. [13] developed a nanoemulsion with lemongrass oil and chitosan, finding an antimicrobial effect that contributes to colour retention and to maintaining the antioxidant activity of fruits. The objective of this study was to evaluate the effect of the application of a nanoemulsion based on orange essential oil and xoconostle extract on the physiological parameters, bioactive compounds and antioxidant activity of avocado fruits (*Persea americana*) during postharvest.

## 2. Materials and Methods

### 2.1. Materials

“Hass” avocado fruits were obtained from the production area of Uruapan, Michoacán, Mexico (19 ° 25’16 “N 102 ° 03’47” W). The xoconostle fruit variety *Opuntia oligacantha* var. Ulapa were from the municipality of Tetepango, Hidalgo, Mexico. The fruits that were used were in a state of physiological maturity.

The following reagents were used in this study: orange essential oil (Reasol, Mexico City, Mexico), soy lecithin (Reasol, Mexico City, Mexico), food-grade mineral oil (UltraSource, Kansas City, USA), catechol (Sigma-Aldrich, St. Louis, Missouri, USA), trichloroacetic acid (TCA) (Sigma-Aldrich, St. Louis, Missouri, USA), formaldehyde (analytical grade reagent), glacial acetic acid (analytical grade reagent), ethanol (analytical grade reagent), methanol (analytical grade reagent), paraplast (Sigma-Aldrich, St. Louis, Missouri, USA), Folin-Ciocalteu reagent (Sigma-Aldrich, St. Louis, Missouri, USA), nitrite sodium (analytical grade reagent), aluminium trichloride (Meyer, Mexico City, Mexico), sodium hydroxide (analytical grade reagent), 2,2-Azino-bis (3-ethylbenzthiazoline-6-sulfonic acid) (ABTS) (Sigma-Aldrich, St. Louis, Missouri, USA), potassium persulfate (analytical grade reagent), and 2,2-diphenyl-1-picrylhydrazyl (DPPH) (Sigma-Aldrich, St. Louis, Missouri, USA).

### 2.2. Preparation of the Nanoemulsion

The nanoemulsion (W/O) was prepared with 70% orange oil (according to Hashtjin and Abbasi [7] the composition of orange essential oil is limonene (94%), myrcene (2%), linalool (0.5%) and some others), 10% xoconostle extract (according to Cenobio-Galindo et al. [5]. The extract has various polyphenols, such as rutin, ferulic acid, quercetin, hydroxybenzoic acid, apigenin, caffeic acid, kaempferol) and 20% soy lecithin. This mixture was sonicated (Sonics Vibra-cell, Connecticut, USA) with a 6-mm-diameter probe for 20 intervals of 50 s of sonication with rest periods of 10 s using 80% amplitude with a frequency of 20 kHz. The obtained nanoemulsion was stored in a refrigerator at 6 °C and protected from light until analysis and use.

#### Determination of the Particle Size and Zeta Potential (ζ)

To verify that the dispersion made was a nanoemulsion, the droplet size was determined, in addition to the stability by means of the zeta potential. The droplet size and zeta potential (ζ) of the system were determined using the methodology developed by Zambrano-Zaragoza et al. [12]; the assessments were made in a Zetasizer Nano-ZS particle size analyzer (Malvern Instruments, Worcestershire, UK), equipped with laser light scattering at an angle of 90°, and the tests were performed in triplicate.

### 2.3. Application of the Nanoemulsion in Avocado “Hass”

Mature fruits homogeneous in weight and size were used and transported under refrigeration conditions (6 °C) to the laboratory located in the city of Tulancingo, Hidalgo, Mexico. The fruits were washed and sanitized with sodium hypochlorite (200 ppm) and were dried at room temperature. the nanoemulsion was applied by spraying the whole fruit. The fruits were coated with concentrated nanoemulsion (CN), nanoemulsion diluted 50% (N50), nanoemulsion 25% (N25) and a control (C) without the addition of nanoemulsion. Dilutions were made with food-grade mineral oil. The fruits were stored at 6 °C and were analyzed every 10 days until day 60.

### 2.4. Weight Loss

The mass was determined by weighing the fruits using a digital scale (OHAUS, Nueva Jersey, USA). The results are expressed as a percentage of weight loss and done in triplicate [14].

### 2.5. Firmness

The method developed by Maftoonazad and Ramaswamy [15] was used by means of a CT3 texture analyzer (Brookfield, Harlow, United Kingdom) adapted with a 5-mm conical strut probe. For the measurement, the epicarp was removed from both sides of the equatorial region. Ten repetitions were made per treatment and the results are expressed in Newtons (N).

### 2.6. pH and Total Soluble Solids

The pH was measured using a pH meter pH 211 microprocessor (Hanna, Rhode Island, USA). For the analysis, 10 g of mesocarp was homogenized with 90 mL of distilled water. The total soluble solids (TSS) was determined with a Pallete PR-101 refractometer (Atago, Washington, USA) and expressed in °Brix [16].

### 2.7. Determination of Polyphenol Oxidase Activity

The activity of polyphenol oxidase (PPO) was determined as reported by Vargas-Ortiz et al. [17]. For each treatment, 5 g of mesocarp was homogenized for 1 min in 15 mL of a 50 mM phosphate buffer (pH 6.5). The mixture centrifuged at 12000 rpm for 30 min at 4 °C and the supernatant used as the enzyme extract. Five hundred μL of 20 mM catechol was added as the substrate with 900 μL of 50 mM phosphate buffer (pH 6.5), and 100 μL of the enzyme extract. For the blank, 500 μL of 10% trichloroacetic acid (TCA) was added. The mixture was incubated for 20 min at 25 °C, and the reaction was stopped with the addition of 500 μL of 10% TCA. The absorbance was read at 410 nm using a spectrophotometer Jenway 6715 UV–Vis (USA). The PPO activity was reported as the increase in absorbance after the reaction time for 100 μL of extract.

### 2.8. Determination of Colour

The external and internal colour was determined using a CM-508d colourimeter (Minolta, Japan) to evaluate the parameters L* (lightness), a* (green to red) and b* (blue to yellow). Five measurements were made on each fruit, and 10 fruits were examined per treatment per day of analysis [15], for internal colour, fruits were cut longitudinally and the same measurements were made as for epicarp.

### 2.9. Bioactive Compounds and Antioxidant Activity

#### 2.9.1. Extraction of Bioactive Compounds

The bioactive compounds of fruit were extracted according Vargas-Ortiz et al. [18]. Five grams of mesocarp was placed in a centrifuge tube, and 15 mL of ethanol/water solution (1:1) was added. The mixture was homogenized for 1 min at 4 °C and then was centrifuged at 12,000 rpm for 5 min at 4 °C. The supernatant was used for the determination of bioactive compounds and antioxidant activity.

#### 2.9.2. Determination of Total Phenols

The total phenolic content was determined by using the Folin–Ciocalteu assay as described by Villa-Rodríguez et al. [19] with some modifications. One milliliter of the extract was mixed with 5 mL of diluted Folin–Ciocalteau reagent (1:10). After 6 min, 4 mL of Na_2_CO_3_ (20%) was added to the mixture, left for 2 h at room temperature and the absorbance against the reagent blank was determined at 760 nm with an UV-Visible spectrophotometer. Total phenolic content was expressed as gallic acid equivalents/100 g of fruit (wet base). All the assays were performed in triplicate.

#### 2.9.3. Determination of Total Flavonoids

Total flavonoid content was measured by the aluminium chloride colorimetric assay using the method of Villa-Rodríguez et al. [19], with some modifications. An aliquot (1 mL) of extracts or standard solutions of quercetin was added to 4 mL of deionized water in a 10 mL flask. To the flask, 300 μL 5% NaNO_2_ was added and after five minutes, 300 μL 10% AlCl_3_. After another five minutes, 2 mL 1M NaOH was added and the volume was made up to 10 mL with deionized water. A blank was prepared in the same manner by using distilled water. The solution was mixed and absorbance was measured against the blank at 415 nm. The total flavonoid content was expressed as mg quercetin equivalents/100 g of fruit (wet base).

#### 2.9.4. Determination of Antioxidant Activity by Inhibiting the DPPH Radical

A 2,2-diphenyl-1-picrylhydrazyl (DPPH) reagent was used to illustrate compounds with antioxidant activity [20]. A solution of DPPH 6.5×10^−5^ M in 80% methanol was prepared, maintaining stirring for 2 h in darkness. Then, 0.5 mL of sample was mixed with 2.5 mL of the DPPH solution, and the mixture was stirred. The absorbance of the mixture was immediately read at a wavelength of 515 nm. The mixture was left to react in darkness for 1 h. As a blank, 80% methanol was used. The results obtained are expressed in mg of ascorbic acid equivalents/100 g of fruit (wet base).

#### 2.9.5. Determination of Antioxidant Activity by Inhibiting the ABTS Radical

Determination of antioxidant activity by inhibiting the radical 2,2-Azino-bis (3-ethylbenzthiazoline-6-sulfonic acid) (ABTS) was performed as described by Vargas-Ortiz et al. [18] with some modifications. A 10 mL solution of 7 mM ABTS was prepared and reacted with 10 mL of 2.45 mM K_2_S_2_O_8_. The mixture was stirred for 16 h in a container in complete darkness. Afterwards the absorbance was measured in a spectrophotometer at 734 nm. The absorbance was adjusted with 20% ethanol to obtain a value of 0.7 ± 0.1. Two hundred μL of the sample was added to 2 mL of ABTS solution and allowed to react for 6 min and absorbance was measured at 734 nm. The results are expressed in mg of ascorbic acid equivalents /100 g of fruit (wet base).

### 2.10. Structural Evaluation of the Epicarp

Visually homogeneous fruits were selected and the fruit epicarp was cut into 1 cm^2^ fragments of the equatorial part. Subsequently, according to the methodology developed by Hernández-Rivero et al. [21], the dissected material was fixed for 24 h in a mixture of formaldehyde, glacial acetic acid, 96% ethanol and distilled water at a ratio of 100:50:50:350. After this time, the tissue was washed with distilled water for 15 min, dehydrated, and infiltrated in an automatic tissue processor TP1020 (Leica, Germany) for 1 h in each of the following solutions: six sequential ethanol solutions at 60%, 70%, 80%, 90%, 96% and 100% and xylene. Then, the tissue was processed in two changes of paraplast solution for 2.5 h in each. The vegetal tissue was embedded in paraplast and cut transversely in 10 slices in a rotary microtome model 820 (Leica, Germany). The slices were mounted on slides, spread on a thermal plate at 25 °C for 24 h and stained with safranin-fast green to show the changes in the lignification of the cell walls, with the greater lignification staining red. The observations were made in an optical microscope model CX31RBSF (Olympus, Japan).

### 2.11. Statistical Analysis

For the statistical analysis, a completely randomized design was used. The results were analyzed with an analysis of variance and when significant differences were observed (*p* < 0.05) between the treatments, the mean comparison was performed by the Tukey method using the NCSS 2007 software (USA).

## 3. Results

### 3.1. Nanoemulsion

The results obtained for the emulsion showed a particle size of 94 ± 8 nm and a zeta potential (ζ) of −106 ± 5 mV, indicating excellent droplet size and stability against phase separation, in addition to translucent appearance, characteristics that are suitable for it to be considered a nanoemulsion [8].

### 3.2. Weight Loss

Figure 1a shows the result obtained for weight loss in avocado fruits. Significant differences (*p* < 0.05) between the treatments were observed. The control (C) fruits lost 9.81 ± 1.93% weight after 10 days as compared with the 2.31 ± 0.18%, 2.13 ± 0.99% and 2.26 ± 0.62% weight loss of fruits coated with nanoemulsion CN, N50 and N25, respectively; maintaining this behavior until the end of the analysis.

### 3.3. Firmness

Significant differences in firmness (*p* < 0.05) between the treatments were observed (Figure 1b) The firmness of the C group decreased from the beginning of the evaluation (6.33 ± 0.81 N to day 10), while the fruits covered with N25 and N50 showed the greatest firmness at day 30 (42.66 ± 4.88 and 39.66 ± 6.40 N, respectively), suggesting that the nanoemulsion is effective in maintaining firmness for longer compared to C.

### 3.4. pH and Total Soluble Solids

From Table 1 it is clear that significant differences in pH between treatments (*p* < 0.05) were found. At day 30, the C group showed a pH of 6.40 ± 0.03, and N25 had a pH of 6.15 ± 0.03, which was the most acid. The results for total soluble solids (Table 1) relate with the pH values, where C had higher total soluble solids than the rest of the treatments at day 30 (4.00 ± 0.01, 2.83 ± 0.05, 2.86 ± 0.05, and 3.23 ± 0.15 for C, CN, N50 and N25, respectively). These results are associated with the pH when indicating the lower acidity in the C group.

### 3.5. Determination of Polyphenol Oxidase Activity

PPO activity (Table 1) showed significant differences (*p* < 0.05) between the treatments. Greater activity in the C group at day 30 (0.29 ± 0.01) was observed, while N25 had the lowest PPO activity (0.21 ± 0.01), observing that the nanoemulsion is effective to keep inactive this enzyme causing the darkening of the mesocarp in fruits.

### 3.6. Determination of Colour

The external color of the avocado during ripening changed from green to black. In Table 2 the results of color are shown. No significant differences (*p* > 0.05) between the treatments were found in the L* parameter, although there is a decrease in value over time. The results of parameter a* (green color) displayed a significant decrease with all treatments after 30 days. The most significant effect was found in the controls (C), indicating a faster maturation which is associated with degradation of chlorophyll. The b* parameter showed significant differences (*p* < 0.05) and decreased in the all treatments after 30 days.

Table 3 shows the color results in the mesocarp of avocado fruits. In parameter L*, there are no significant differences (*p* > 0.05) between the treatments for each day of analysis, but a decrease in the brightness values respect to time is observed. Differences in the parameter a* (*p* < 0.05) are observed for treatments at day 30. C showed a greater increase in values, which is associated with the loss of green color, behavior similar to that observed for parameter b *, where C showed a loss of yellow color compared to the rest of the treatments at day 30.

### 3.7. Bioactive Compounds and Antioxidant Activity

#### 3.7.1. Total Phenols

The results demonstrate significant differences (*p* < 0.05) in total phenolic compounds in the avocado mesocarp over a 60-day period as well as between controls and treatments with nanoemulsions (Table 4). After 30 days, there was a substantial drop in total phenolic concentration for all the treatments, but afterwards, a significant increase followed until day 60. The highest concentrations of total phenols at day 60 were found with N25 and N50 treatments (214.29 ± 7.78 and 240.15 ± 16.29 mg EAG/100 g).

#### 3.7.2. Total Flavonoid Content

Table 4 shows an increase in the total flavonoid concentration with respect to time. At day 60, the highest concentration of flavonoids was observed in the N25 and N50 treatment groups (48.18 ± 1.78 and 47.77 ± 2.82 mg EQ/100 g, respectively).

#### 3.7.3. Determination of Antioxidant Activity

There were significant differences in the antioxidant activity when the DPPH radical was inhibited (*p* < 0.05) (Table 4). The treatments coated with the nanoemulsion presented an increase in antioxidant activity from day 30 to 60 that corresponded to the increase in bioactive compounds (phenols and flavonoids). The C treatment showed a different behavior, whereby there was a decrease in antioxidant activity with respect to time (Table 4). The results for the antioxidant activity in avocado mesocarp with the inhibition of the ABTS radical are shown in Table 4. These results are similar to those for the inhibition of the DPPH radical. Treatments coated with the nanoemulsion presented an increase in the antioxidant capacity from day 30 to day 60. The N25 treatment presented the best values of antioxidant activity at days 30 and 60 (211.40 ± 5.15 and 233.87 ± 5.94 mg EAA/100 g, respectively), while the C treatment presented the lowest values (112.48 ± 13.77 and 68.32 ± 5.42 mg EAA/100 g).

### 3.8. Structural Evaluation

In Figure 2, the changes in the pericarp of the avocados with respect to time are shown, demonstrating that the exocarp is constructed by epidermal cells in a vertical disposition. In the immature fruit (day 0) a thick cuticle covering the epidermal tissue is clearly visible. One of the most prominent changes of the treatments up to day 30 was the lignification of the cell wall as accessed through the red staining protocol by the action of safranin.

## 4. Discussion

Postharvest weight loss is mainly attributed to perspiration caused by a deficit of steam pressure of the product in relation to its environment [22]. Sellamuthu et al. [23] found that weight loss decreased by applying thyme oil vapour and packaging of avocados from different cultivars in modified atmosphere and Russo et al. [24] observed that the firmness of the fruits decreased during storage from the fifth day in avocado fruits stored in different atmospheres (CO_2_/O_2_). Maftoonazad and Ramaswamy [15] applied a coating based on methylcellulose in avocado fruits, observing that coated treatments managed to maintain avocado firmness, which indicates that the use of a nanoemulsion as a coating is an effective alternative to prevent loss of firmness

Higher pH values can be associated with the conversion of organic acids and other complex molecules present in fruits into sugars, which is a source of energy reserve used in the metabolic processes during ripening [24,25]. Saucedo-Pompa et al. [26] coated avocado fruits with candelilla wax and ellagic acid and found that the fruits that were not coated had pH values closer to neutrality compared with the other treatments. Aguirre-Joya et al. [16] observed similar results in avocados coated with candelilla wax, pectin, aloe mucilage and polyphenols of *Larrea tridentate*, where the control showed a tendency to increase °Brix value as the storage time increased.

PPO is responsible for the darkening of the avocado, and it has been reported that PPO activity increases with a more accelerated ripening in stored fruits [23,27]. The results obtained in the present study are similar to those reported by Tesfay and Magwaza [2] in which the activity of PPO in the mesocarp of avocados decreased by applying edible moringa leaf extract coatings to avocado, showing that the maintenance of the integrity of the membrane in the coated fruit contributed to the reduction of darkening.

A decrease in luminosity in the epicarp of avocado has been related to the synthesis of some anthocyanins such as cyanidin 3-O-glucoside that confers dark colors [19]. The treatments that were coated with the nanoemulsion showed slower changes in colour, directly affecting the ripening attributes of the fruits [28]. Villa-Rodríguez et al. [19] evaluated the changes during the ripening of avocado fruits, reporting that there is a change in colour with respect to time that is related to the degradation (mainly chlorophyll) and synthesis of compounds. Correa-Pacheco et al. [29] determined color in mesocarp and epicarp of avocado fruits coated with chitosan-thyme essential oil nanoparticles, observing that there were no differences between the coated and uncoated treatments; however, the results of the present study show that the nanoemulsion was able to maintain the color of the mesocarp as opposed to the fruits that were not coated. Cenobio-Galindo et al. [5] mentioned that bioactive compounds from xoconostle can remain effectively included in certain coatings without losing their activity, finding that certain compounds, such as rutin, ferulic acid, quercetin and apigenin, were maintained even after being incorporated into films and provided antioxidant activity.

Tesfay et al. [30] mentioned that the production of polyphenols can be stimulated by some biotic or abiotic factors; therefore, the application of the nanoemulsion could act as stimulus to increase phenolic compounds. Taking into account the results obtained, it is possible that this behavior is related to the diluted nanoemulsion treatments forming a more stable semipermeable coating with respect to the CN, and the stress in the fruit can remain for a longer time, affecting the synthesis of compounds. Wang et al. [20] evaluated the concentration of total phenols in different cultivars of avocado, finding that the cultivar “Hass” had high concentrations in the seeds and peels. Similar results were found by Vinha et al. [25], in which the concentration of total phenolics in avocados from the Algarve region in Portugal were highest in the seed and skin.

Sellamuthu et al. [23] indicated that essential oils can act as signaling compounds, causing a slight stress situation in the fruit, resulting in the increase in certain compounds, such as flavonoids. The procyanidins (catechin and epicatechin) are the main flavonoids present in avocados and are considered potent antioxidants with beneficial health effects [19,20]. From the results of Vinha et al. [25], it was clear that the highest concentration of total flavonoids was detected in the pericarp and seeds. Their reported value in the mesocarp was lower at 21.9 ± 1.0 mg / 100 g than our findings, in which, even at day 0, a higher amount (37.07 ± 0.71 mg EQ/100 g) was noted.

The antioxidant activity present in the avocado is given by the bioactive compounds present in the fruit, which include phenolic compounds such as procyanidins, chlorophylls and carotenoids [15]. Wang et al. [20] determined the antioxidant activity in different cultivars of avocado by inhibiting the DPPH radical and found that the cultivar “Hass” had the greatest antioxidant activity. Villa-Rodríguez et al. [19] determined antioxidant activity in avocado with different extractions and found that lipophilic extracts showed a greater antioxidant activity than hydrophilic extracts, compared by the different methods analyzed. There are reports suggesting that essential oils acting as signaling compounds are also capable of increasing antioxidant activity; therefore, nanoemulsions can fulfil this function [31].

Higher lignification of the cell walls in fruits treated with C and CN, compared to the N25 and N50 treatments, is an indication of accelerated maturation of the fruit. The colour change among treatments is because the green colour prevails in cells that have a slower metabolism, and cells that showed a more lignified wall were stained red by the action of safranin [32]. The effects that retard the fruit maturation (weight loss, firmness and PPO activity) are attributed to the action of the nanoemulsion as a coating. In all the treatments, it was observed that the cuticle remained until the end of the analysis. Schroeder [33] mentioned that the elimination or destruction of this protective epidermal layer makes the surface of the fruit more susceptible to attack by fungal and bacterial infections or can cause physiological disorders as a result of drying out.

## 5. Conclusions

The incorporation of bioactive compounds with antioxidant activity through a nanoemulsion based on orange essential oil and xoconostle extract as a coating on avocado fruits had beneficial effects on postharvest storage of the fruit. The application of the nanoemulsion decreased the percentage of weight loss, retained firmness, decreased enzymatic activity, delayed its darkening and decreased effects on epidermal cells. These properties of the nanoemulsion translate into slower ripening of the avocados compared to the control. The nanoemulsion coating can be considered as an alternative treatment to increase the postharvest life of the avocado fruit.

## Figures and Tables

**Figure 1 antioxidants-08-00500-f001:**
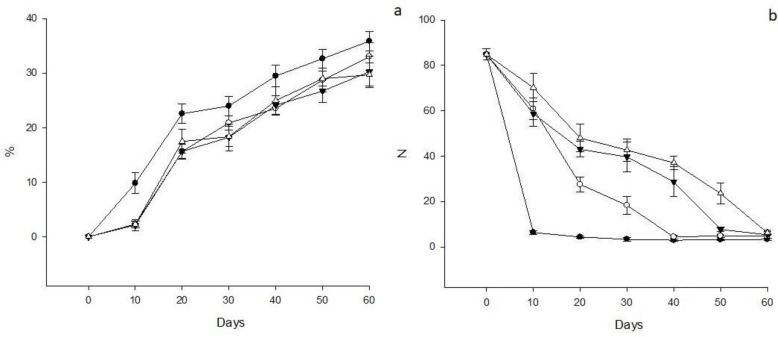
(**a**) Result for the percentage of weight loss in avocado. (**b**) Result for firmness in avocado. The results are expressed as means ± standard deviation. N = Newtons. ● = C, ▼ = N25, △ = N50, ○ = CN.

**Figure 2 antioxidants-08-00500-f002:**
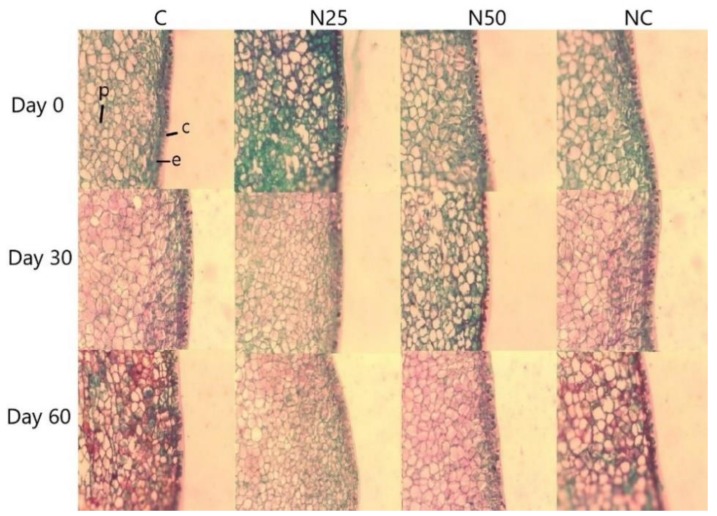
Structural evaluation of the avocado epicarp with respect to time. The columns indicate the same treatment with respect to time and the rows indicate the same day of analysis for different treatments, c = cuticle, e = epidermis, p = parenchyma.

**Table 1 antioxidants-08-00500-t001:** Physicochemical changes in avocado during postharvest.

Days/Treatments	C	N25	N50	CN
**pH**				
0	5.62 ± 0.06^aA^	5.62 ± 0.04^aA^	5.63 ± 0.04^aA^	5.60 ± 0.09^aA^
30	6.40 ± 0.03^cB^	6.15 ± 0.03^aB^	6.33 ± 0.02^bB^	6.33 ± 0.04^bB^
60	6.56 ± 0.05^aC^	6.70 ± 0.07^bC^	6.70 ± 0.01^bC^	6.72 ± 0.01^bC^
**TSS**				
0	1.13 ± 0.05^aA^	1.03 ± 0.05^aA^	1.10 ± 0.10^aA^	1.00 ± 0.01^aA^
30	4.00 ± 0.01^cB^	2.83 ± 0.04^aB^	2.86 ± 0.05^aB^	3.23 ± 0.15^bB^
60	4.06 ± 0.11^aB^	4.03 ± 0.05^aC^	4.00 ± 0.07^aC^	4.16 ± 0.05^aC^
**PPO activity**				
0	0.21 ± 0.01^aA^	0.23 ± 0.01^aA^	0.21 ± 0.01^aA^	0.23 ± 0.01^aA^
30	0.29 ± 0.01^cB^	0.21 ± 0.01^aA^	0.24 ± 0.00^bB^	0.23 ± 0.01^bA^
60	0.33 ± 0.01^cC^	0.26 ± 0.01^aB^	0.26 ± 0.02^aB^	0.31 ± 0.01^bB^

Different lowercase letters in the same row indicate significant differences (*p* < 0.05) between treatments at same analysis day. Different capital letters in the same column indicate significant differences (*p* < 0.05) between each treatment at different analysis day. Total soluble solids (TSS) are expressed in °Brix. Polyphenol oxidase (PPO) activity is expressed in increase in absorbance/0.1 mL of extract.

**Table 2 antioxidants-08-00500-t002:** Color changes in the skin of the avocado during postharvest.

Days/Treatments	C	N25	N50	CN
**L***				
0	35.78 ± 1.70^aA^	35.09 ± 2.18^aA^	34.57 ± 1.52^aA^	35.01 ± 1.47^aA^
30	29.08 ± 1.81^aB^	28.78 ± 1.66^aB^	29.72 ± 1.05^aB^	29.57 ± 1.13^aB^
60	22.66 ± 0.67^aC^	21.52 ± 0.85^aC^	23.03 ± 0.29^aC^	22.45 ± 0.29^aC^
**a***				
0	−7.38 ± 0.84^aA^	−7.24 ± 0.60^aA^	−7.11 ± 0.73^aA^	−6.90 ± 1.53^aA^
30	2.05 ± 0.32^cB^	−0.25 ± 1.15^aB^	−0.94 ± 1.57^aB^	1.40 ± 0.42^bB^
60	4.51 ± 0.30^aC^	4.11 ± 0.38^aC^	4.99 ± 0.50^aC^	4.43 ± 0.41^aC^
**b***				
0	29.78 ± 2.65^aA^	29.83 ± 2.24^aA^	30.11 ± 1.81^aA^	30.08 ± 2.31^aA^
30	25.11 ± 0.57^aB^	26.68 ± 0.56^aB^	22.25 ± 0.39^bB^	19.53 ± 1.59^cB^
60	25.39 ± 0.64^aB^	25.44 ± 1.07^aB^	23.41 ± 1.53^abB^	21.76 ± 0.43^bB^

Different lowercase letters in the same row indicate significant differences (*p* < 0.05) between treatments at same analysis day. Different capital letters in the same column indicate significant differences (*p* < 0.05) between each treatment at different analysis day. L* = lightness, a* = green to red, b* = blue to yellow.

**Table 3 antioxidants-08-00500-t003:** Color changes in the mesocarp of the avocado during postharvest.

Days/Treatments	C	N25	N50	CN
**L***				
0	77.42 ± 1.95^aA^	77.32 ± 0.78^aA^	76.59 ± 1.24^aA^	75.99 ± 1.64^aA^
30	69.73 ± 0.70^aB^	71.43 ± 1.09^aB^	71.31 ± 0.49^aB^	69.60 ± 1.00^aB^
60	59.94 ± 2.15^aC^	59.94 ± 2.15^aC^	62.32 ± 1.80^aC^	60.12 ± 0.93^aC^
**a***				
0	−7.67 ± 0.58^aA^	−7.99 ± 0.63^aA^	−7.95 ± 0.50^aA^	−9.05 ± 0.77^aA^
30	−2.38 ± 0.89^bB^	−4.78 ± 0.47^aB^	−4.26 ± 0.40^aB^	−4.26 ± 0.40^aB^
60	3.50 ± 0.29^bC^	1.61 ± 0.23^aC^	1.30 ± 0.34^aC^	1.57 ± 0.19^aC^
**b***				
0	45.50 ± 0.97^aA^	45.70 ± 1.02^aA^	44.65 ± 0.86^aA^	47.98 ± 0.47^aA^
30	41.96 ± 0.76^bB^	44.67 ± 1.36^aA^	44.75 ± 0.78^aA^	45.57 ± 1.84^aA^
60	33.77 ± 0.44^bC^	35.73 ± 0.59^aB^	35.95 ± 1.48^abB^	33.85 ± 0.58^bB^

Different lowercase letters in the same row indicate significant differences (*p* < 0.05) between treatments at same analysis day. Different capital letters in the same column indicate significant differences (*p* < 0.05) between each treatment at different analysis days. L* = lightness, a* = green to red, b* = blue to yellow.

**Table 4 antioxidants-08-00500-t004:** Bioactive compounds and antioxidant activity changes in the mesocarp during postharvest.

Days/Treatments	C	N25	N50	CN
**Total phenols**				
0	247.33 ± 2.77^aA^	238.14 ± 17.11^aA^	244.46 ± 7.05^aA^	239.00 ± 11.70^aA^
30	122.07 ± 4.56^bC^	160.85 ± 2.28^aB^	127.24 ± 6.50^bB^	120.34 ± 5.65^bC^
60	152.23 ± 6.03^cB^	214.29 ± 7.78^bA^	240.15 ± 16.29^aA^	164.87 ± 5.54^cB^
**Total flavonoids**				
0	37.07 ± 0.71^aB^	36.25 ± 0.94^aC^	36.46 ± 0.35^aB^	36.04 ± 0.61^aB^
30	32.34 ± 1.23^cC^	44.27 ± 0.35^aB^	36.04 ± 2.22^bB^	46.74 ± 2.49^aA^
60	42.63 ± 2.33^bA^	48.18 ± 1.78^aA^	47.77 ± 2.82^abA^	44.27 ± 1.55^bA^
**DPPH**				
0	462.15 ± 25.61^aA^	483.38 ± 15.13^aA^	477.14 ± 32.29^aA^	469.23 ± 25.59^aA^
30	247.84 ± 10.91^bB^	309.43 ± 17.04^aC^	242.85 ± 20.33^bB^	239.52 ± 7.52^bC^
60	179.18 ± 17.22^cC^	435.52 ± 13.69^aB^	439.68 ± 18.47^aA^	318.17 ± 13.75^bB^
**ABTS**				
0	228.44 ± 7.79^aA^	225.60 ± 6.34^aA^	223.28 ± 4.71^aB^	221.73 ± 13.41^aA^
30	112.48 ± 13.77^bB^	211.40 ± 5.15^aB^	112.48 ± 6.05^bC^	128.75 ± 12.46^bB^
60	68.32 ± 5.42^dC^	233.87 ± 5.94^bA^	310.57 ± 5.15^aA^	195.13 ± 14.33^cA^

Different lowercase letters in the same row indicate significant differences (*p* < 0.05) between treatments at same analysis day. The different capital letters in the same column indicate significant differences (*p* < 0.05) between each treatment at different analysis days. Total phenols are expressed in mg GAE/100 g, total flavonoids are expressed in mg QE/100 g, DPPH are expressed in mg AAE/100 g, and ABTS are expressed in mg AAE/100 g.

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
