# Peer review of "Influence of Bioactive Compounds Incorporated in a Nanoemulsion as Coating on Avocado Fruits (Persea americana) during Postharvest Storage: Antioxidant Activity, Physicochemical Changes and Structural Evaluation"

_antioxidants, 2019, doi:10.3390/antiox8100500_

Round 1
Reviewer 1 Report
This is an interesting manuscript describing a study looking at extend the postharvest storage of avocado fruit. Overall, the methods used are appropriate for the study. The use of a nanoemulsion to extend shelf life is novel. There are few minor typographical errors that the authors should address. These are listed below by line number.
Line 87, page 2: "11." should be deleted.
Line 135, page 4: Sodium carbonate needs subscript numbers.
Line 143, page 4: Sodium nitrite and aluminum chloride need subscript numbers.
Line 213, Table 1: The authors should consider defining TSS and PPO in the legend of the table to remind readers of the terms.
Line 287, page 9: Carbon dioxide and oxygen need subscript numbers.
Reviewer 2 Report
I found this paper very interesting and valuable. Some of the parts should be classified:
1. In all part of manuscript please use the same convention of unit, e.g. mL or ml;
2. Authors used orange essential oil (Reasol, Mexico). No data about detail composition of essential oil, that was used in experiment. According to the best of our knowledge, the composition of essential could be differ depending of fraction, fruit oil;
3. Authors used for preparation of nanoemulsion xoconostle extract. Authors determined total polyphenols, total flavonoids. If possible, please add HPLC or LC-MS or NMR profile of fracion;
4. Total phenols are expressed in mg GAE/100 g. Please precise if it is on dry or wet base.
5. DPPH are expressed in mg AAE/100 270 g, ABTS are expressed in mg AAE/100 g. Please precise if it is on dry or wet base.
6. What is "absolute xylene"? Please precise.
7. Correct Figure 1, I mean 60 day on the chart.
8. How authors explain changes in total phenols for N25 and N50 as well as CN ? (I mean fluctuated values);
